# Emergent Structures and Lifetime Structure Evolution in Artificial Neural Networks

Siavash Golkar
Flatiron Institute
New York University
sgolkar@flatironinstitute.org

## ABSTRACT

Motivated by the flexibility of biological neural networks whose connectivity structure changes significantly during their lifetime, we introduce the Unrestricted Recursive Network (URN) and demonstrate that it can exhibit similar flexibility during training via gradient descent. We show empirically that many of the different neural network structures commonly used in practice today (including fully connected, locally connected and residual networks of different depths and widths) can emerge dynamically from the same URN. These different structures can be derived using gradient descent on a single general loss function where the structure of the data and the relative strengths of various regulator terms determine the structure of the emergent network. We show that this loss function and the regulators arise naturally when considering the symmetries of the network as well as the geometric properties of the input data.

## 1 INTRODUCTION

A remarkable property of biological neural netowrks (BNNs) is their adaptability in the face of new environments, different tasks and when coping with structure damage [1]. In contrast, despite their successes, artificial neural networks (ANNs) are limited in their applicability, and structures need to be designed for each particular task. Inspired by the *genetic evolution* of BNNs, an active field of neural architecture search has emerged leading to specialized networks which excel at specific tasks [3]. However, there is no ANN analog of the *lifetime evolution* of the structure of BNNs which provide flexibility in the face of new challenges or damage.

The question therefore naturally arises of 1. whether there exist flexible ANNs which can adapt their connectivity structure to the task they are trained on *during their lifetime* and 2. whether there exists a new machine learning paradigm based on these flexible networks which can compete with the highly specialized networks in use today. The present work is a small step towards answering some of these questions. In particular, we introduce the Unrestricted Recursive Network (URN), and show that when trained end to end via stochastic gradient descent, a URN dynamically *chooses* its structure. Specifically, depending on the geometric structure of the data and the choice of regulator hyperparameters, the same URN can turn into networks which are recursive or feedforward, fully connected or locally connected (as in CNNs), and can choose whether or not to have residual skip connections. We also show that the specific form of the URN and the loss function used is mostly determined by various symmetry arguments.

### Related work

Dynamical network architectures where perviously discussed in Fahlman and Lebiere [4], where the network is grown one neural at a time. More recently, it was shown that recurrent neural networks with linear and convolutional layers can improve performance in specific circumstances [2, 7].

## 2 EMERGENT STRUCTURES

### Motivation

We start the discussion by the simple observation that (almost) any network architecture can be embedded in a recursive network.

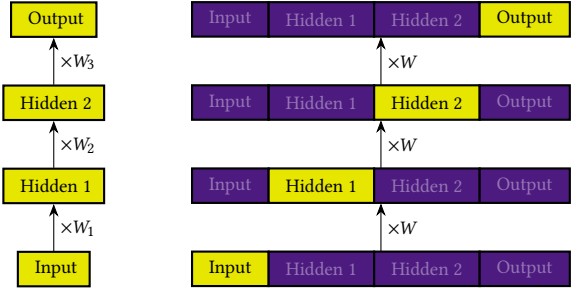

(a) Original network.                    (b) Equivalent structure.

**Figure 1: Embedding a multi-layer perceptron (a) in an unrolled unstructured recursive network (b). The purple nodes denote neurons that are identically zero.**

For demonstration, let us consider a feed-forward neural network with two hidden layers. Fig. 1a shows a cartoon of this network with $W_{i=1, 2, 3}$ denoting the weights of each layer. We can embed this feed-forward architecture inside a larger recursive structure by concatenating the neurons of all layers as in Fig. 1b and similarly embedding the weights of the different layers inside a larger weight matrix $W$ defined as (biases are treated similarly):

$$W = \begin{pmatrix} 0 & 0 & 0 & 0 \\ W_1 & 0 & 0 & 0 \\ 0 & W_2 & 0 & 0 \\ 0 & 0 & W_3 & 0. \end{pmatrix} \quad (1)$$

It is simple to verify that consecutively applying the $W$ matrix (along with the activation function) 3 times as in Fig. 1b is equivalent to the original MLP network. This block sub-diagonal structure of the weight matrix is a signature of MLPs. In a similar manner, almost all feed-forward neural networks can be embedded in recursive networks.

### The Unstructured Recursive Network

We now show that it is possible to perform the converse of the above demonstration: starting from a general recursive structure

without layers, we can arrive at an emergent network which can be interpreted as a feedforward multi-layered perceptron.

Consider a learning task with $d_{in}$ and $d_{out}$ specifying the dimension of the vectorized input and output. Motivated by the embedding arguments of the previous section, we define a network with as little structure as possible as follows. First, we embed the input data in the first $d_{in}$ elements of $N^i$, a vector of length $S$ representing all the neurons of the system:

$$N_i^{(0)} = \begin{cases} x_i & i \leq d_{in} \\ 0 & i > d_{in} \end{cases}, \tag{2}$$

where $x_i$ denotes the $i$'th component of each vectorized training sample and the superscript (0) denotes that this is the input of the network (zeroth iteration). We then define define a discreet iterative update rule for processing the data:

$$N_i^{(l+1)} = \phi(W_{ij}N_j^{(l)} + b_i), \tag{3}$$

where $W$ is an $S \times S$ matrix, $b$ is the bias, and $\phi$ is the non-linear activation function. Note that $W$ is initialized as a (He normal) dense matrix and does not have the block structure of Eq. 1 at the beginning of training. We apply this update rule a total of $I$ times and then read off the output as the final $d_{out}$ nodes of the neurons.

$$\hat{y}_i = N_{S-i}^{(I)}, \; i \leq d_{out} \tag{4}$$

A cartoon of this structure is given in Fig. 2. We call this architecture the Unstructured Recursive Network (URN). The only structural hyperparameters here are the total number of neurons $S$ and the number of iterations $I$. We will see that neither of these parameters are indicative of the structure of the final emergent network.

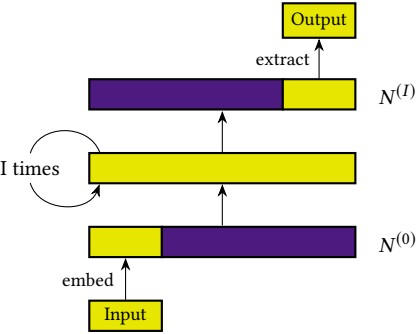

**Figure 2: Schematic of the Unstructured Recurrent Network.**

We train this network using standard loss functions and gradient descent methods. Specifically, we demonstrate our methodology on classification tasks using a multi-class cross-entropy loss function with added $L_1$ regulators on both network weights as well as the neuron activations after each iteration of the update rule. Correspondingly we have two hyperparameters $c_W$ and $c_N$ which control the strengths of these regulator terms:

$$L = \frac{1}{N}\sum L_{XE}(y, \hat{y}) + c_W|W| + c_N \sum_{l=1}^{I} |N^{(l)}|. \tag{5}$$

The use of these regulators is to promote sparsity in the number of active neurons and nonzero weights of the network, which in turn make the emergent structure simpler to interpret.

## Emergent structures

A primary finding of this paper is that when training a URN on a classification task with high values of weight and activity regulation, the topology of the emergent network is a feed-forward multi-layer perceptron. As an example we train a URN with a total of $S = 5000$ neurons and $I = 4$ iterations on a binary classification task comprised of distinguishing inputs sampled from two uniform concentric 10-d spherical shell distributions. Fig. 3 shows a generic result for neural activities and the weight matrix for a network trained on 4000 training samples for 200 epochs with Adam Optimizer and learning rate $7 \times 10^{-4}$, $c_W = 5 \times 10^{-7}$ and $c_N = 2 \times 10^{-5}$ (Because of the simplicity of the task in this section, we only choose hyperparameters which consistently lead to 100% test accuracy). For these plots, we have discarded the inactive neurons (i.e. neurons with zero activation) and sorted the remaining neurons according to the iteration number at which they are first activated.

Of the 5000 total initial neurons, only $123 \pm 15$ (mean + STD across 5 trials) neurons remain active at the end of training. Comparing Fig. 3 to the weight and neural activity structure of the previous section (see Eq. 1 and Fig. 1b), we see that the neurons have neatly organized into an MLP with 3 hidden layers comprised of $115 \pm 15$, $4 \pm 1.4$, and $3 \pm 0.7$ neurons. In order to ascertain that this structure is indeed the correct topology of the emergent network, we manually set all other weights of the network to zero and verify there is no change in the network output empirically. For a video of the evolution of a URN during training see https://youtu.be/hvlAnwW-IyY.

## URN with emergent number of layers

In the previous section, the number of layers of any emergent MLP is by construction equal to $I$, the number of the iterations of the recursive network. This is necessarily true since there are $I$ applications of the update rule (Eq. 3) in the derivation of the final values of the output (Eq. 4). In order to relax this requirement, we need to allow for pathways in the computation graph of the output values which include different numbers of the update rule. The network can then choose dynamically how many 'layers' to utilize. This can be done using residual connections either on the input or on the output nodes.

*Residual output nodes.* Consider the following update rule:

$$N_i^{(l+1)} = \begin{cases} \phi(W_{ij}N_j^{(l)} + b_i) & i < S - d_{out} \\ N_i^{(l)} + \phi(W_{ij}N_j^{(l)} + b_i) & i \geq S - d_{out} \end{cases}. \tag{6}$$

This modification has the simple interpretation that it allows for the output of the neural network to accumulate gradually in the output nodes. The network can therefore dynamically cut off further changes to the output after iteration $L \leq I$. The number of layers of the emergent network would then effectively be $L$. Fig. 4 depicts the results of the training of a URN on the same problem as in the previous section (Fig. 3), with the modified update rule in Eq. 6. The emergent network now has one hidden layer (compared to three in the previous section) despite the number of iterations $I$ being 4. This can be attributed to the (lack of) complexity of

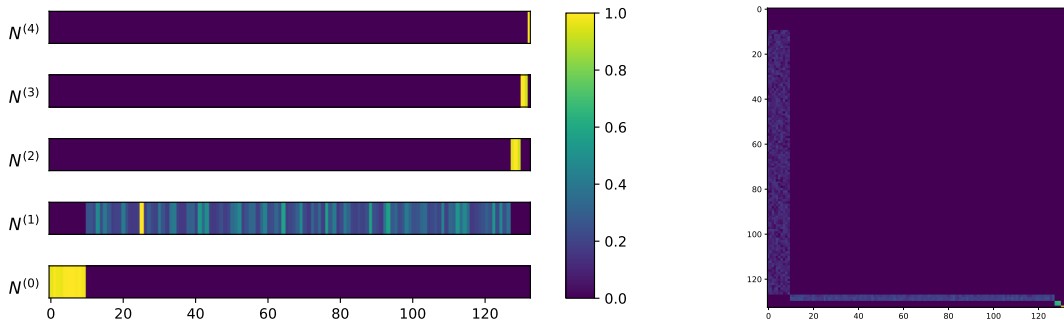

**Figure 3: Activities (left) and weight matrix (right) of a URN of total size $S = 5000$ and $I = 4$ iterations on the concentric sphere dataset with $d = 10$. The weight matrix and activities of neurons exhibit an emergent MLP structure.**

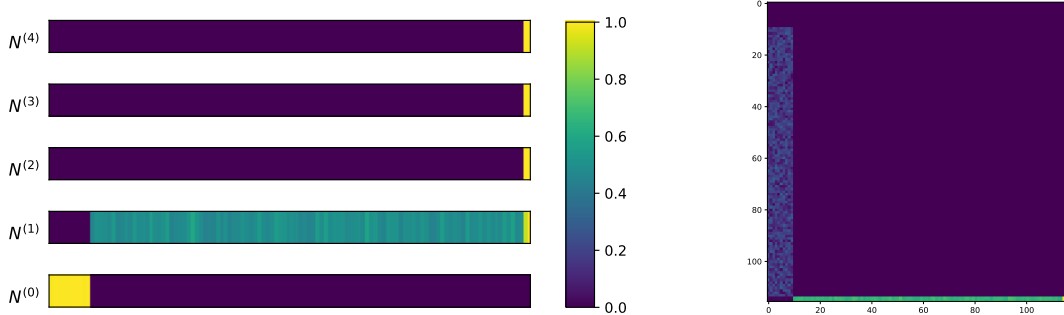

**Figure 4: Same setting as Fig. 3 with output nodes which have a residual update rule (Eq. 6).**

the dataset which becomes linearly separable after one layer. This observation is further reinforced in Sec. 3 where in a more difficult problem (classification on CIFAR-10) the emergent network utilizes the maximum allowed number of layers (i.e. $L = I$).

*Residual input nodes.* This alternative modification has the intuitive interpretation of continuously feeding the input into the input nodes at every step of the iteration:

$$N_i^{(l+1)} = \begin{cases} x_i + \phi(W_{ij}N_j^{(l)} + b_i) & i \leq d_{in} \\ \phi(W_{ij}N_j^{(l)} + b_i) & i > d_{in} \end{cases}. \qquad (7)$$

The implementation of residual connections on the input nodes has the advantage that it allows for the formation of skip connections in the emergent network. However, because of this mixing of different layers (i.e. neurons that have different numbers of iterations of the update rule applied), it also leads to neural activity patterns that are harder to interpret in terms of a simple feed-forward network. We leave the analysis of the emergence of more complicated networks with skip connections and feedback loops to future work.

## 3 INCORPORATING INPUT STRUCTURE

In this section we discuss the circumstances under which convolutional NNs (CNN) or rather locally connected networks (LCN) which are CNNs without weight sharing can arise from a URN. In an LCN, the neurons of each layer are connected to a small neighborhood of neurons in the previous layer. The definition of this neighborhood implicitly requires a proximity or a distance measure defined on the neurons or on the individual components of

the input. Consequently, a dataset which has no such proximity information, or more generally datasets which are invariant under permutation of the components of the input (such as the spheres in Sec. 2), combined with an update rule which preserves this symmetry (e.g. Eq. 3), would generally lead to emergent structures which also respect this permutation symmetry and hence do not have local connectivity structure. This permutation symmetry is naturally broken in many tasks. Here, we specialize to image recognition as an example and show how this symmetry breaking naturally leads to emergent networks with local connectivity structure.

Let us assume that the input of the network is a two-dimensional matrix of size $d_x \times d_y$. Implicit in this notation is the assumption of a Euclidean metric which determines the relative distance of different pixels on the 2D plane (the argument also applies to curved or other non-trivial geometries). It is therefore natural that when we embed this metric space inside the larger structure of the URN, there will be also be a metric induced on this larger space given by the uplift of the 2D Euclidean metric of the input. The simplest such metric would be a product metric where there is a single extra dimension perpendicular to the nodes assigned as the input (see Fig. 5):

$$ds^2 = ds_{\text{input}}^2 + \beta dz^2, \qquad (8)$$

where $\beta$ is a hyperparameter determining the perpendicular length scale compared to the directions parallel to the input.

This induced geometric structure on the neurons of the network allows us to add extra regulator terms to the loss function which are interpretable as penalizing the synaptic length connecting different

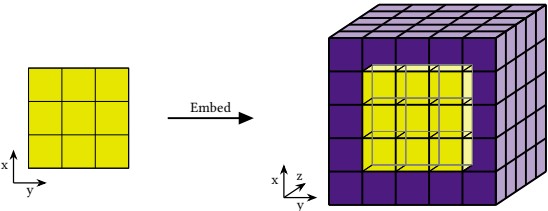

**Figure 5: Embedding an input with geometric structure in the neurons of the URN. A metric on the input samples (left) naturally induces a metric structure on this larger space (right).**

neurons. This term was not allowed under the permutation symmetry of the previous section. Also, this term is still invariant under the parts of the permutation symmetry which remain unbroken after the addition of the metric structure (e.g. discrete 90 degree rotations).

$$L = \frac{1}{N} \sum L_{\text{XE}}(y, \hat{y}) + c_W|W| + c_N \sum_{l=1}^{I} |N^{(l)}| + c_{\text{syn}} \sum_{i<j} |W_{ij}| d_{ij}^{\gamma}, \quad (9)$$

where $c_W$ and $c_N$ are as before, $d_{ij}$ is the distance of the $i$'th and $j$'th neuron under the metric in Eq. 8, $\gamma$ is the distance power hyperparameter, and $c_{\text{len}}$ determines the strength of the new regulator term penalizing the length of each synapse.

We performed an experiment using monochromatic CIFAR-10 images using a URN with an uplift geometry equivalent to a $x \times y \times z$ cube of $60 \times 60 \times 6 = 21,600$ total neurons (see Fig. 5). The input embedding rule (Eq. 2) is modified as follows: the $32 \times 32$ inputs are expanded to $60 \times 60$ using interpolation and are embedded in the $z = 1$ neurons. We use $I = 4$ iterations of the residual output update rule described in Eq. 6. Finally, the center 10 neurons at $z = 6$ or designated as the output nodes. The emergent network structure is depicted in Fig. 6, where forward going weights (connecting smaller $z$ neurons to larger $z$ neurons) are depicted in red, backward going weights in green and equal $z$ weights are depicted in black. A clear feedforward and locally connected structure is apparent with a very few green/black weights. Without hyperparameter fine-tuning this network achieves a test accuracy of 52%, a 10% improvement over the same structure with only weight and activity regulators.

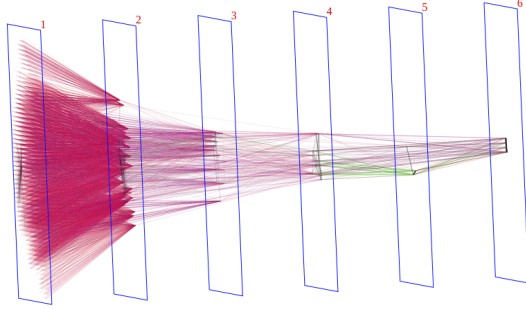

**Figure 6: Connectomics of a URN trained on CIFAR-10 with synaptic length regularization.**

## 4 DISCUSSION

In this paper we have an empirical demonstration that many of the neural network structures in use today can dynamically emerge from the same general framework of the URN. We showed in examples that the final topology of these networks is easily interpretable as feed-forward MLPs with number of layers and number of neurons per layer determined during training. Furthermore, we showed that given input data with proximity information (e.g. a metric), we can naturally extend the URN loss function such that we can derive locally connected networks whose generalization performance is considerably improved. These demonstrations, however, are only the first stages of this project and much work remains to be done. For example, one can ask, how does the emergent network topology vary with task difficulty. This question is currently under study and beyond the scope of the demonstrations in this paper. One can also ask many other questions: e.g. under what circumstances, if any, recurrent neural NNs emerge from a URN or if it is possible to somehow naturally incorporate weight sharing such that we can arrive at a convolutional network. Finally, a theoretical understanding of why we generically arrive at feed-forward networks beyond simple intuitive arguments is still needed.

*Never-Ending Structure Accumulation.* In light of the recent works in continual and never-ending learning [6], and to circle back to the points raised in the introduction, we propose the following alternative learning scheme. Let us assume that we are given a series of related tasks of gradually increasing difficulty. For example, in vision, these can start from simple edge detection and end with image classification. We can intuitively predict what will happen if we train a network with dynamically chosen architecture on these tasks consecutively using a compatible lifelong learning algorithm which minimizes performance loss on prior tasks. When trained on the simple tasks, the emergent network would be shallow with few layers. However, as more complex tasks are trained, the depth of the network would grow and each consecutive tasks would naturally build on top of the structures already present in the architecture. Preliminary results show that this expectation is borne out when training a URN in conjunction with the lifelong learning algorithm from Golkar et al. [5] on a series of simple to difficult image tasks.

This line of argument and experiments suggest an alternative learning paradigm to today's highly specialized networks specifically built for each task. In this learning paradigm, which we dub Never-Ending Structure Accumulation or NESA, the structure is simply determined by the series of simple to difficult tasks which culminate in the final ML problem of interest. The responsibility of the ML practitioner in NESA would then be to design this series tasks. While this is not a trivial undertaking for many ML problems, it brings the problem of training ANNs much closer to how BNNs learn to perform new tasks during their lifetime.

*Acknowledgments.* We would like to thank Jack Hidary, Kyunghyun Cho, Cristina Savin, Owen Marschall, Yann LeCun, Dmitri Chklovskii and Anirvan Sengupta for interesting discussions and input. This work is partly supported by the James Arthur Postdoctoral Fellowship.

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
