# OpenReview forum: "Emergent Structures and Lifetime Structure Evolution in Artificial Neural Networks"
_NeurIPS.cc/2019/Workshop/Neuro_AI — Real Neurons & Hidden Units @ NeurIPS 2019 Poster_

### Official Review · AnonReviewer2 · 2019-09-24
**promising new approach for joint architecture+weight learning by reducing generic connectivity matrix**

**Clarity:** 5

**Comment:**

Questions
1. In Fig. 2, why can the URN not be collapsed to a single layer with I iterations, where the first d_in neurons always deliver input (and not just once; in e.g. vision the brain also receives continuous input), and the last d_out neurons always deliver output (neuro-analogy of responding early/adaptive inference)? This approach might lead to some changes with training (continuous "image on"; computing loss at all times/only the last step), but seems like a more generic and more brain-like implementation.
2. How are c_W and c_N chosen? The text stated using "hyperparameters which consistently lead to 100% test accuracy" -- are c_W=5x10^-7 and c_N=2x10^-5 the only ones for which that worked? It would be cool to see which sparsity constraints lead to promising models and if those tell us anything about the sparsity in the brain.

Suggestions
1. I only gave this a 4 and not a 5 because there is no baseline/comparison to other methods. The URN approach should be compared to existing architecture/weight search/evolution approaches to determine if/how it differs and to tell us which models are better than others.
2. The premise of this work for me is a more flexible search space than most architecture search approach which are restricted to the operations that are defined a priori. For instance, current architecture search techniques could not find local convolutions unless it's already part of the search space. It would be great to see if you can make the space even more flexible (cf. Q.1; maybe starting from an all-to-all connectivity matrix that already includes skip/recurrent connections by definition) and scale it up to the ImageNet dataset. The resulting model could then be tested on its performance and match to brain (using publicly available data, e.g. www.Brain-Score.org)
3. I realize this is particularly hard in this context, but is there any relevant developmental bio data that you could compare to? Perhaps developmental tracing studies which you compare the URN evolution to?


**Category:**

Common question to both AI & Neuro

**Clarity Comment:**

The paper is very well-written.

Some minor things are confusing in the manuscript:
1. page 2, "outpud" should be "output"
2. page 4, "embeded" --> "embedded"
3. page 4, "classificaiton" --> "classification"
4. page 4, "Anon [?]" reference missing
5. I find "Section 2" clearer than "§2" (and "§3")

**Evaluation:**

4: Very good

**Importance:**

4: Very important

**Importance Comment:**

While Machine Learning has produced high-performing architectures (both manually and with architecture search methods), it is still unclear how basic connectivity patterns could emerge in natural and artificial systems.
This paper addresses this question by starting from an all-encompassing, iteratively applied weight matrix which is reduced to consecutive operations by training on the task with a sparsity constraint.

**Intersection:**

3: Medium

**Intersection Comment:**

Emergent structures are certainly a highly relevant topic in both Machine Learning and Neuroscience.
For this workshop in particular, the paper is motivated by biological structural changes over a network's lifetime, but doesn't compare their method to any biological data. That said, I still think the work bears a very relevant approach and is developed with a neuro/bio perspective. The bio context might also allow this approach to further develop without having to chase SOTA.

**Rigor Comment:**

The visualizations and analyses are convincing.
However, the method is not compared to alternative approaches such as RL-based or evolution-based architecture search (the most closely related paper is probably Pham et al. 2018, https://arxiv.org/abs/1802.03268). It is thus hard to judge whether the approach shown here will find different networks than other methods or find them more efficiently.


**Technical Rigor:**

3: Convincing

---

### Official Review · AnonReviewer1 · 2019-09-26
**A nice empirical work on the neural architecture**

**Clarity:** 4

**Comment:**

I wonder why the intrinsic geometric structure of the CIFAR-10 images is not enough to induce the local connectivity. Also, it would be interesting to check, when the network is trained from the learned connectivity structure but with randomized weights, how fast the network reaches convergence compared to the original unstructured model.

**Category:**

Common question to both AI & Neuro

**Clarity Comment:**

The manuscript is clearly written, but this was achieved by using a very small font size.

**Evaluation:**

3: Good

**Importance:**

4: Very important

**Importance Comment:**

In this work, the author(s) trained dense recurrent neural networks on simple classification tasks, then found that feedforward structures naturally emerge in the networks. I believe this line of work will provides insights on why the brain is filled with feedback and recurrent connections when feedforward DNN is sufficient for solving tasks that the brain faces, though the presented work is still a small number of empirical simulations.

**Intersection:**

2: Low

**Intersection Comment:**

The axonal projection patterns in the brain is arguably regulated innately, and mostly fixed after the developmental period. Thus, the presented work is less relevant to the brain, compared to the traditional genetic algorithm-based approach. Still, the geometry-based regularization introduced in this work is a potentially interesting intersection with the neural architecture in the brain.

**Rigor Comment:**

On "this is necessarily true since..." in p2:
Although it is indeed trivially true that the number of layers is upper bounded by the number of iterations, it does not necessarily provide the lower bound. In particular, in the presence of recurrent connections, a shallow network with recursive working memory should be a valid solution too.

**Technical Rigor:**

4: Very convincing

---

### Decision · Program_Chairs · 2019-10-02

Accept (Poster)